# The Effect of a Nucleation Layer on Morphology and Grain Size in MOCVD-Grown β-Ga_2_O_3_ Thin Films on C-Plane Sapphire

**DOI:** 10.3390/ma15238362

**Published:** 2022-11-24

**Authors:** Lauris Dimitrocenko, Gundars Strikis, Boris Polyakov, Liga Bikse, Sven Oras, Edgars Butanovs

**Affiliations:** 1Institute of Solid State Physics, University of Latvia, Kengaraga Street 8, LV-1063 Riga, Latvia; 2Institute of Physics, University of Tartu, W. Ostwaldi Str. 1, 50412 Tartu, Estonia; 3Institute of Technology, University of Tartu, Nooruse 1, 50411 Tartu, Estonia

**Keywords:** gallium oxide, MOCVD, thin films, UWBG materials, nucleation layer

## Abstract

β-Ga_2_O_3_ thin films grown on widely available c-plane sapphire substrates typically exhibit structural defects due to significant lattice and thermal expansion mismatch, which hinder the use of such films in electronic devices. In this work, we studied the impact of a nucleation layer on MOCVD-grown β-Ga_2_O_3_ thin film structure and morphology on a c-plane sapphire substrate. The structure and morphology of the films were investigated by X-ray diffraction, atomic force microscopy, transmission and scanning electron microscopy, while the composition was confirmed by X-ray photoelectron spectroscopy and micro-Raman spectroscopy. It was observed that the use of a nucleation layer significantly increases the grain size in the films in comparison to the films without, particularly in the samples in which H_2_O was used alongside O_2_ as the oxygen source for the nucleation layer growth. Our study demonstrates that a nucleation layer can play a critical role in obtaining high quality β-Ga_2_O_3_ thin films on c-plane sapphire.

## 1. Introduction

Gallium oxide (Ga_2_O_3_) has recently attracted significant scientific attention as a new, promising ultra-wide bandgap semiconductor [1,2]. Several Ga_2_O_3_ polymorphs exist of which monoclinic β-Ga_2_O_3_ is the thermodynamically most stable phase [3]. It has a 4.6–4.9 eV wide bandgap [4]; therefore, the theoretical limit value of its breakdown voltage exceeds those of the more commonly studied SiC and GaN [5]. Together with its high mechanical strength and chemical and thermal stability [4], β-Ga_2_O_3_ properties show promise for a variety of applications in power electronics [5,6,7] and solar-blind deep-ultraviolet photodetectors [8,9].

β-Ga_2_O_3_ thin films have been grown by a variety of methods, such as pulsed laser deposition [10,11] and magnetron sputtering [12]; however, molecular beam epitaxy [13,14] and metalorganic chemical vapour deposition (MOCVD) [15,16,17,18] are the most promising methods for high deposition rates and high-quality device fabrication. MOCVD in particular has the advantage of high growth rates (up to 10 μm/h) with sub-nanometre surface roughness [16]. Typically, triethylgallium (TEGa) is used as the Ga precursor and O_2_ as the oxygen precursor in Ga_2_O_3_ MOCVD growth; however, films with a comparable quality have also been grown with the significantly cheaper trimethylgallium (TMGa) [19,20]. Regarding the oxygen precursor, commonly used O_2_ is the established choice, yet the use of N_2_O can lead to a lower background charge carrier concentration [21] and the use of water (H_2_O) vapour can potentially have a positive effect on the kinetic conditions during film growth due to the presence of hydrogen [20]. There are only a few papers reporting the use of H_2_O as an oxygen precursor; Alema et al. demonstrated a reduction in carrier concentration by adding a small amount of H_2_O to the O_2_ flow during the film growth [22], while Lee et al. and Chen et al. reported Ga_2_O_3_ film growth using H_2_O as the oxygen precursor without further discussions on the choice [23,24,25].

The use of native Ga_2_O_3_ substrates leads to the highest quality epitaxial films [10,21]; however, they are still expensive and thus currently uneconomical for large-scale production. On the other hand, α-Al_2_O_3_ (sapphire) substrates are widely available, but the significant 6.6% lattice and thermal expansion mismatch between the (0001) c-plane sapphire and β-Ga_2_O_3_ [26] leads to structural defects during the growth process such as threading dislocations, which hinder the use of such films in electronic devices. Large numbers of threading dislocations lead to formation of low-angle grain boundaries represented as a configuration of edge dislocations, whose axes lie in the plane of the boundary between the grains. In the case of GaN on c-plane sapphire, the high density of dislocations due to the large (~14%) lattice mismatch was overcome by introducing a buffer layer through controlled nucleation of highly crystalline islands [27]. As for Ga_2_O_3_, to the best of our knowledge, there are no dedicated reports on the role of a nucleation or a buffer layer in MOCVD growth. Alema et al. reported deposition of a 30 nm thick low-temperature (650 °C) buffer layer on c-plane sapphire before the film overgrowth in order to improve its crystalline quality [16], and Chen et al. used a 15 nm thick nucleation layer, deposited at 600 °C, to grow ε-Ga_2_O_3_ film on c-plane sapphire [28]. However, no comparison of the films grown with and without a nucleation layer has been made. A positive effect of a seed layer was recently demonstrated by Gu et al. [29] for sputter-deposited Ga_2_O_3_ films on variously oriented silicon substrates, as well as of α-Cr_2_O_3_ buffers for α-Ga_2_O_3_ growth by Polyakov et al. [30].

In this work, we investigated and compared MOCVD-grown β-Ga_2_O_3_ thin films (A) without any nucleation layer, (B) with a nucleation layer grown by using an O_2_ precursor as the oxygen source and (C) with a nucleation layer grown by using a combination of H_2_O and O_2_ precursors. The focus of this study was the film structure and morphology. It was observed that the use of a nucleation layer significantly increases the grain size in our films, particularly in the samples in which H_2_O was used alongside O_2_ in the nucleation layer growth. Further development of a more complex nucleation or buffer layer could lead to significantly higher crystalline quality β-Ga_2_O_3_ films on widely available c-plane sapphire wafers.

## 2. Materials and Methods

Heteroepitaxial β-Ga_2_O_3_ thin films were grown on 2-inch c-plane sapphire wafers (Biotain Crystal Co. Ltd., Xiamen, China) using the AIX-200RF (AIXTRON, Herzogenrath, Germany) MOCVD system. Trimethyl gallium (TMGa) was used as the gallium source, while water (H_2_O) and oxygen (O_2_) were used as oxygen sources. Nitrogen (N_2_) was used as a carrier gas with the total flow of 3000 standard cubic centimetres per minute (sccm) during the deposition process. Before growth, the sapphire substrates were annealed for 2 min at 1070 °C in a N_2_ atmosphere at 200 mbar pressure. The main Ga_2_O_3_ layer for all samples was deposited in a N_2_ atmosphere and 30 mbar pressure at 820 °C temperature. TMGa and O_2_ flow rates were 120 µmol/min and 20 mmol/min, respectively. For a 30 min growth process the thickness of the as-grown thin films was measured to be around 1.5 μm.

Three types of growth procedures were implemented and studied:Sample A was prepared without any nucleation layer.Sample B was prepared with a low-temperature (LT) nucleation layer deposited using O_2_ precursor as the oxygen source. The amorphous LT layer (around 30 nm thick) was deposited at 720 °C for 150 s with a 90 µmol/min TMGa flow rate and a 16 mmol/min O_2_ flow rate (Step 1 in Figure 1). Then, the sample temperature was raised to 1030 °C to perform high-temperature annealing of the layer (Step 2 in Figure 1), supposedly crystallizing it as suggested by X-ray diffraction measurements in Appendix A. After a 150-s-long dwell time, the temperature of the sample was decreased to the main layer growth temperature (see above) followed by the 30 min deposition (Step 3 in Figure 1).Sample C was prepared with a nucleation layer deposited using both precursors H_2_O and O_2_ as the oxygen sources. One LT layer was grown as it was described for Sample B. An additional LT layer was overgrown on top of the first one for 60 s using TMGa and H_2_O as the gallium and oxygen sources, respectively. The flow rates were 90 µmol/min for TMGa and 320 µmol/min for H_2_O precursors. Before the growth of the main layer, the temperature was increased to anneal the LT layers similarly as for Sample B.

The thin film surface morphology was examined with a scanning electron microscope (SEM, Thermo Scientific Helios 5 UX, Waltham, MA, USA) and an atomic force microscope (AFM, Veeco, Dimension Edge, Plainview, NY, USA) in tapping mode using a Pointprobe non-contact AFM tip NCHR-50, while the film thickness was determined by ellipsometry (Woollam RC2-XI, Lincoln, NE, USA). X-ray diffraction (XRD, PANalytical, X’Pert Pro, Malvern, UK) patterns were measured using Cu Kα irradiation in Bragg–Brentano Theta–Theta geometry, with a step size of 0.026 deg, tube powered by 40 kV and 30 mA and rocking curves (ω-scans) were measured using parallel beam optics and monochromatic (Cu Kα_1_) radiation, with a step size of 0.001 deg. The lamellae were prepared by focused ion beam (FIB, Thermo Scientific Helios 5 UX, Waltham, MA, USA) to investigate the cross-section of the thin film with a transmission electron microscope (TEM, FEI Tecnai G20, Hillsboro, OR, USA) operated at 200 kV. The thin film was coated with a 30 nm thick Au layer and a 1.5 µm thick Pt layer to protect the surface from Ga^+^ exposure during FIB preparation. Micro-Raman spectroscopy measurements were performed using a TriVista 777 confocal Raman system (Princeton Instruments, Trenton, NJ, USA, 750 mm focal length, 1800 lines/mm grating) equipped with an Olympus microscope and UIS2 MPlanN 100 ×/0.90 objective, a continuous-wave diode-pumped laser Cobolt Samba 150 (HÜBNER Photonics, Kassel, Germany) (λ = 532 nm) and Andor iDus DV420A-OE CCD camera (Andor Technology, Belfast, Northern Ireland). The X-ray photoelectron spectrometry (XPS) measurements were performed by an ESCALAB 250Xi (ThermoFisher, Waltham, MA, USA) instrument to determine the film chemical composition. An Al Kα X-ray tube with the energy of 1486 eV was used as an excitation source, the size of the analysed sample area was 650 μm × 100 μm and the angle between the analyser and the sample surface was 90°. An electron gun was used to perform charge compensation. The base pressure during the spectra acquisition was better than 10^−7^ mbar.

## 3. Results and Discussion

The surface morphology of the as-deposited Ga_2_O_3_ thin films was investigated with AFM and SEM and is shown in Figure 2. It can be observed that the size of the crystalline domains or grains significantly increases when a nucleation layer is used. The sample without any nucleation layer (Sample A) exhibits a small-grained (sub-micron) structure without any visually distinguishable orientation, while the use of a nucleation layer in Sample B and Sample C seems to lead to much larger grains. Furthermore, the use of a H_2_O precursor alongside O_2_ in the nucleation layer growth in Sample C increases the grain size in the overgrown Ga_2_O_3_ film even further. During the establishment of the film growth process, a series of films using various O_2_ and H_2_O precursor ratios were deposited and it was observed that increasing the proportion of O_2_ in the precursor flow leads to a higher growth rate and larger grains (see SEM images in Appendix A); however, it also favours growth of grains of crystalline orientations other than (−201), as can be seen in the XRD patterns in Appendix A. This is caused by the intensive parasitic reaction between TMGa and O_2_ precursors. In order to reduce the influence of the parasitic reaction, a low process pressure (below 50 mBar) was used. By increasing the pressure above 50 mBar, the film deposition rate decreases drastically. However, in the case of the H_2_O precursor, even 200 mBar pressure during growth does not affect the deposition rate. Therefore, the use of H_2_O during the nucleation layer growth in Sample C seems to reduce the misorientation of grains in the initial growth step, which consequently leads to fewer structural defects in the overgrown film and thus, larger grains. The mean grain size was calculated from processing several AFM measurements for each sample. Regarding the surface roughness, maximum peak height values, R_p_, were determined from the AFM measurements and calculated to be 178.2 nm (Sample A), 202.9 nm (Sample B) and 367.6 nm (Sample C). The increase in the surface roughness can be attributed to the formation of larger single-crystalline grains, which tend to protrude.

The crystalline structure of the prepared films was studied with XRD and TEM. A typical XRD pattern (2Θ scan) of the films can be seen in Figure 3a, showing a highly crystalline monoclinic β-Ga_2_O_3_ phase (ICDD-PDF #41-1103) oriented along [−201] direction. Since all the samples gave qualitatively similar patterns, only one is shown here. Rocking curves (ω-scans) of the (−201) peak for all prepared samples showed similar full-width half-maximum (FWHM) values in the range of 1.82–1.93°, which is comparable to the best reported values in the literature for β-Ga_2_O_3_ growth on c-plane sapphire [31,32]. Furthermore, lamellas were prepared in SEM-FIB in order to perform the film cross-section studies with TEM and reveal the crystalline structure of the films. The Ga_2_O_3_/sapphire interface can be seen in Figure 3b, confirming the oriented growth of the film. Interplanar distance (d-spacing) of the resolved atomic planes in the Ga_2_O_3_ film was measured to be 4.73 Å, which matches well with the [−201] growth direction [33]. The analysis of the selected area electron diffraction (SAED) pattern in Figure 3c also supports this. SEM images of the cross-sections (see Figure 3d for Sample C and Appendix A for Samples A and B) show columnar structures with different contrast, which could be visible due to slightly misoriented grains. Grain boundaries visible in the cross-section cuts (see Figure 3e) are typically low-angle grain boundaries with misalignment of 1–3°; however, regions of grain misalignment of up to 18° could be occasionally identified in Sample A and Sample B (see Appendix A). The characteristic quantities for the film structure and morphology are summarised in Table 1.

The film composition and purity were verified with micro-Raman spectroscopy and XPS. The Raman spectrum of a Ga_2_O_3_ film is shown in Figure 4a together with a pure sapphire substrate spectrum for comparison. The sapphire substrate peaks are marked with ‘S’, while the rest can be attributed to multiple phonon modes with A_g_ and B_g_ symmetry of β-Ga_2_O_3_, closely matching the spectral positions reported previously in the literature [34]. The XPS survey scan (see Figure 4b) only indicated the presence of Ga and O elements in stoichiometry closely matching the Ga_2_O_3_ compound, as well as surface carbon contaminants. The spectra were calibrated to the adventitious C1s peak at 285.0 eV binding energy. No other elements were detected. The high-resolution spectra of Ga3d and O1s peaks are shown in the inset. The Ga3d peak was located at 20.0 eV with an overlapping O2s peak at around 23.5 eV, while the O1s peak was measured to be at 530.8 eV with a shoulder contribution from the C-O chemical state in the organic surface contaminations. The Ga and O peak binding energy values correspond to the Ga_2_O_3_ compound as expected [35].

## 4. Conclusions

In this study, we demonstrated an improvement in grain size and morphology of MOCVD-grown β-Ga_2_O_3_ thin films on c-plane sapphire by using a nucleation layer in the initial film growth step. The structure and morphology of the films were investigated by XRD, TEM, SEM and AFM, and the composition was confirmed by XPS and micro-Raman spectroscopy. The films in which a H_2_O precursor was used alongside O_2_ as the oxygen source in the nucleation layer growth exhibit larger grains and slightly better crystallinity in comparison to the films with an O_2_ only nucleation layer. This is possibly due to less intensive parasitic reactions between TMGa and H_2_O in comparison to TMGa and O_2_ precursors, causing fewer structural defects in the nucleation layer and thus larger and highly oriented crystalline grains in the overgrown β-Ga_2_O_3_ film. Our study demonstrates that a nucleation layer can play a critical role in obtaining high quality β-Ga_2_O_3_ thin films on c-plane sapphire, therefore further development of a more complex nucleation or buffer layer recipe could lead to significantly higher crystalline quality β-Ga_2_O_3_ films on widely available c-plane sapphire substrates.

## Figures and Tables

**Figure 1 materials-15-08362-f001:**
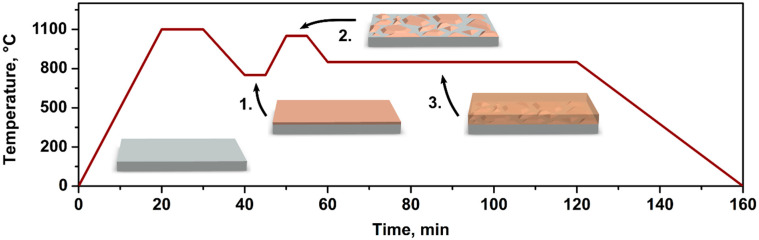
A schematic of Ga_2_O_3_ thin film MOCVD growth steps using a nucleation layer. After the high-temperature annealing of the substrate, a few-nm thick low-temperature Ga_2_O_3_ layer is deposited at 720 °C (Step 1). It is then annealed for 150 s at 1030 °C (Step 2), followed by the overgrowth of Ga_2_O_3_ film at 820 °C (Step 3).

**Figure 2 materials-15-08362-f002:**
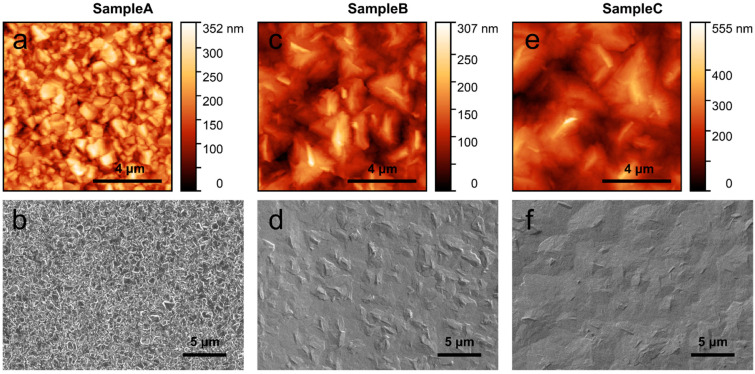
Atomic force microscopy (10 µm × 10 µm scans) and scanning electron microscopy images of (**a**,**b**) Sample A, (**c**,**d**) Sample B and (**e**,**f**) Sample C, respectively, showing the surface morphology of the as-grown Ga_2_O_3_ films.

**Figure 3 materials-15-08362-f003:**
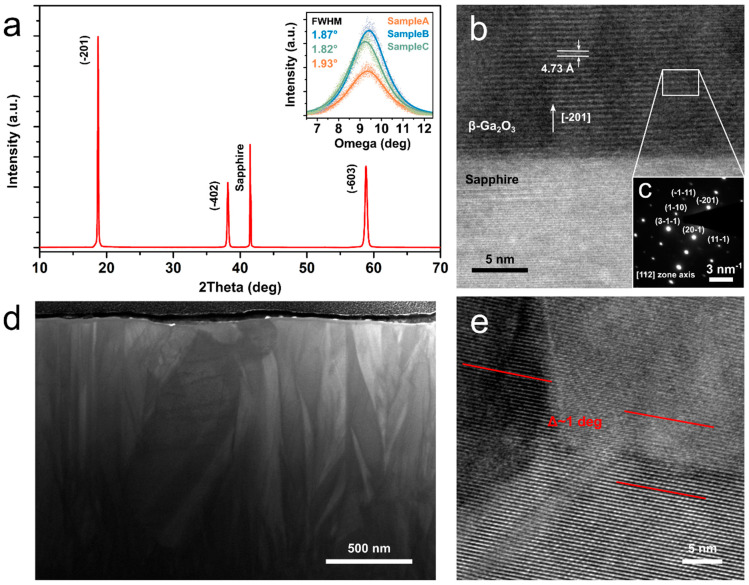
(**a**) A typical X-ray diffraction pattern of a Ga_2_O_3_ film on c-plane sapphire, all samples gave qualitatively similar patterns. The inset shows rocking curves (ω-scans) of the (−201) peak of the studied samples. (**b**) A transmission electron microscope image of the Ga_2_O_3_/sapphire interface in Sample C, showing the Ga_2_O_3_ growth direction and interplanar spacing, with (**c**) selected area electron diffraction pattern of the Ga_2_O_3_ film. (**d**) Scanning electron image of the cross-section cut of Sample C. (**e**) An example of a low-angle grain boundary typically observed in Sample C.

**Figure 4 materials-15-08362-f004:**
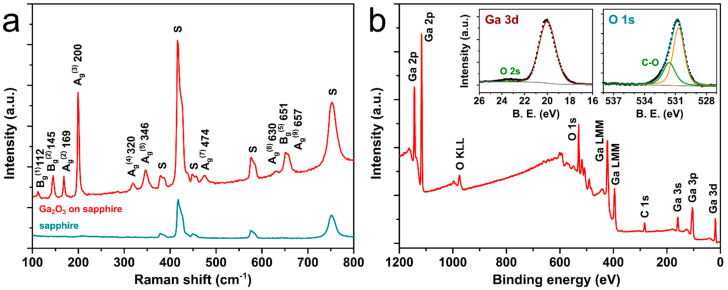
(**a**) A typical micro-Raman spectrum of as-grown Ga_2_O_3_ thin films, the phonon modes and their respective values in cm^−1^ are indicated above each peak. The sapphire substrate spectrum is shown for comparison, its Raman peaks are marked with ‘S’. (**b**) A typical XPS survey spectrum of as-grown Ga_2_O_3_ thin films. The insets show individual Ga 3d and O 1s peak scans. All the prepared samples gave qualitatively similar results.

**Table 1 materials-15-08362-t001:** Characteristic parameters for the as-grown β-Ga_2_O_3_ film structure and morphology.

	Sample A	Sample B	Sample C
Maximum peak height (roughness) R_p_, nm	178.2	202.9	367.6
Mean grain size, μm	0.389	1.502	1.989
Rocking curve FWHM, degrees	1.93	1.87	1.82

## Data Availability

The data supporting this study’s findings are available from the corresponding author upon reasonable request.

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
