# Peer review of "The Effect of a Nucleation Layer on Morphology and Grain Size in MOCVD-Grown β-Ga2O3 Thin Films on C-Plane Sapphire"

_materials, 2022, doi:10.3390/ma15238362_

Round 1

Reviewer 1 Report

The authors present a concise, detailed, and clear report of the effect of a nucleation layer for the heteroepitaxial growth of β-Ga2O3 on a sapphire substrate. The addition of an altered nucleation layer to the epitaxial stack appears to improve the surface morphology. The authors plausibly show that this nucleation layer increases grain size, and might have an affect on subgrain disorder induced low angle grain boundaries. This is a valuable and impactful step towards improved heteroepitaxial crystal quality, and these improvements to grain size and surface morphology are valuable to the epitaxy research community. Future work will be needed to clarify whether this approach will be effective to significantly improve crystalline quality. 

The authors seem to assume, a priori, that the nucleation layer reduces dislocation density, instead of showing it to be true as a matter of experimental result. This strikes this reviewer as inappropriate. Therefore, the authors must remove claims about improved dislocation density from the manuscript, in the opinion of this reviewer. No direct experimental evidence of reduced dislocation density is given. It is suggested that the authors revise the manuscript to focus on the results of improved grain size and morphology, rather than speculating about dislocation density. 

Recommendation is to accept for publication with minor but mandatory revisions, listed below: 

1.     A more descriptive title along the lines of “The effect of nucleation layer on surface morphology…” might be more helpful for readers. The current title suggests a more in-depth study including theoretical explanation of the “role” of a nucleation layer which is lacking in this study.

2.     The authors refer to the 150s anneal of the LT-nucleation layer as a “crystallization”, which seems to be conjecture. What evidence do the authors have that the main outcome of this anneal is to “crystallize” the nucleation layer? In the analogous case of GaN/sapphire, there are a number of theoretical and experimental investigations including growth-interrupt studies with detailed TEM, in-situ studies, and first-principles computations showing the role of the anneal and the LT-NL, but no evidence of such effect is shown here. The authors should either show evidence that some crystalline evolution (coalescence? Grain growth/coarsening? Change in crystal structure, analogous to the cubic/wurtzite GaN transition?) is taking place—or, clarify that these descriptions of “crystallization” are conjectures based on prior literature, which must be cited. 

3.     The Figure caption for Fig. 1 might be revised. The phrase: “It is then annealed for 150 seconds at 1030°C in order to create highly-crystalline and epitaxially oriented nucleation islands” strikes this reviewer as speculative, and not really backed up by the results in this report. Even if it were proven in this report, a figure caption is not the right place to put such claims. What evidence do the authors have that there are any “highly-crystalline and epitaxially oriented” islands? Based on the XRD and TEM it seems clear that the islands are not particularly well oriented and have a significant amount of tilt and/or twist disorder. 

4.     The TEM images shown in the manuscript are not conclusively proving any overall crystal quality improvement. The best way to show crystal quality improvement is with a plan-view, scattering contrast image to show dislocations. The only other measure of dislocation density, the rocking-curve width, shows negligible difference between samples. Therefore, the authors are encouraged to either add more evidence and discussion to prove their claim of improved crystal quality, or revise the manuscript to remove such claims. 

5.     What is the importance of Fig. 3c? A high-magnification image like this gives little information regarding crystal quality, since it shows only a small volume. The authors only mention the figure once in the text, just to mention that the measured lattice spacing is consistent with the crystallographic orientation already known from XRD. This reviewer would suggest removing this figure or giving a more detailed explanation of its importance in the text. 

6.     Would the authors consider including Figure S3 in the main text? That figure strikes this reviewer as much more useful in evaluating the overall heteroepitaxial quality of these films than Fig 3c. 

Author Response

Reviewer 1

The authors present a concise, detailed, and clear report of the effect of a nucleation layer for the heteroepitaxial growth of β-Ga2O3 on a sapphire substrate. The addition of an altered nucleation layer to the epitaxial stack appears to improve the surface morphology. The authors plausibly show that this nucleation layer increases grain size, and might have an affect on subgrain disorder induced low angle grain boundaries. This is a valuable and impactful step towards improved heteroepitaxial crystal quality, and these improvements to grain size and surface morphology are valuable to the epitaxy research community. Future work will be needed to clarify whether this approach will be effective to significantly improve crystalline quality. 

The authors seem to assume, a priori, that the nucleation layer reduces dislocation density, instead of showing it to be true as a matter of experimental result. This strikes this reviewer as inappropriate. Therefore, the authors must remove claims about improved dislocation density from the manuscript, in the opinion of this reviewer. No direct experimental evidence of reduced dislocation density is given. It is suggested that the authors revise the manuscript to focus on the results of improved grain size and morphology, rather than speculating about dislocation density. 

Recommendation is to accept for publication with minor but mandatory revisions, listed below: 

  1. A more descriptive title along the lines of “The effect of nucleation layer on surface morphology…” might be more helpful for readers. The current title suggests a more in-depth study including theoretical explanation of the “role” of a nucleation layer which is lacking in this study.

The title of the manuscript has been changed accordingly. The new title is “The effect of a nucleation layer on morphology and grain size in MOCVD grown β-Ga2O3 thin films on c-plane sapphire”.

  1. The authors refer to the 150s anneal of the LT-nucleation layer as a “crystallization”, which seems to be conjecture. What evidence do the authors have that the main outcome of this anneal is to “crystallize” the nucleation layer? In the analogous case of GaN/sapphire, there are a number of theoretical and experimental investigations including growth-interrupt studies with detailed TEM, in-situ studies, and first-principles computations showing the role of the anneal and the LT-NL, but no evidence of such effect is shown here. The authors should either show evidence that some crystalline evolution (coalescence? Grain growth/coarsening? Change in crystal structure, analogous to the cubic/wurtzite GaN transition?) is taking place—or, clarify that these descriptions of “crystallization” are conjectures based on prior literature, which must be cited. 

All the unsupported claims related to the processes occurring to the nucleation layer during the high-temperature annealing have been removed from the manuscript. A short discussion on crystallization has been added and supported by a new Fig.S1, which shows XRD patterns of the layers after annealing. The patterns show β-Ga2O3 Bragg peaks in contrast to the X-ray amorphous LT-NL.

  1. The Figure caption for Fig. 1 might be revised. The phrase: “It is then annealed for 150 seconds at 1030°C in order to create highly-crystalline and epitaxially oriented nucleation islands” strikes this reviewer as speculative, and not really backed up by the results in this report. Even if it were proven in this report, a figure caption is not the right place to put such claims. What evidence do the authors have that there are any “highly-crystalline and epitaxially oriented” islands? Based on the XRD and TEM it seems clear that the islands are not particularly well oriented and have a significant amount of tilt and/or twist disorder.

We agree that the statement in the Fig.1 caption was too speculative and has now been removed.

  1. The TEM images shown in the manuscript are not conclusively proving any overall crystal quality improvement. The best way to show crystal quality improvement is with a plan-view, scattering contrast image to show dislocations. The only other measure of dislocation density, the rocking-curve width, shows negligible difference between samples. Therefore, the authors are encouraged to either add more evidence and discussion to prove their claim of improved crystal quality, or revise the manuscript to remove such claims.

We have now removed the unsupported claims on the improved crystalline quality and shifted the focus of the manuscript on the improved grain size and morphology.

  1. What is the importance of Fig. 3c? A high-magnification image like this gives little information regarding crystal quality, since it shows only a small volume. The authors only mention the figure once in the text, just to mention that the measured lattice spacing is consistent with the crystallographic orientation already known from XRD. This reviewer would suggest removing this figure or giving a more detailed explanation of its importance in the text.

Fig.3(b) and (c) are intended to support the XRD results, as well as to show the sapphire/Ga2O3 interface, which indicates a smooth and oriented crystalline transition from the substrate to the nucleation layer. SAED is needed to confirm the measurements in the TEM image since they can be inconclusive on their own. Furthermore, we have now added extra SEM and TEM images to the figure as was suggested by you in Question 6, which now makes it more informative.

  1. Would the authors consider including Figure S3 in the main text? That figure strikes this reviewer as much more useful in evaluating the overall heteroepitaxial quality of these films than Fig 3c.

Images (c) and (d) from Figure S3 have now been added to Figure 3 in the main text, as well as the corresponding changes in the text have been made.

Reviewer 2 Report

The manuscript is devoted to the development of a method for improving the structural quality of heteroepitaxial gallium oxide films using buffer layers. The results are obtained using an comprehensive number of methods. The manuscript is well structured. The results are described clearly. The obtained results are important for the development of power and sensor electronics devices based on gallium oxide. Before accepting this manuscript, the following comments should be taken into account:

1. Buffer layers to increase structural quality, reduce dislocation density were also used for α-Ga2O3 grown https://iopscience.iop.org/article/10.1088/1361-6463/ac962f/meta

2. You write: "... we demonstrated an increase of grain size and crystallinity of MOCVD-grown β-Ga2O3 thin films on c-plane sapphire by using a nucleation layer, which helps to reduce the dislocation density in the initial film..." I did not find estimations of dislocation densities in the text of manuscript.

3. I recommend presenting a table for samples A, B and C with a comparison of their structural properties.

Author Response

Reviewer 2

The manuscript is devoted to the development of a method for improving the structural quality of heteroepitaxial gallium oxide films using buffer layers. The results are obtained using an comprehensive number of methods. The manuscript is well structured. The results are described clearly. The obtained results are important for the development of power and sensor electronics devices based on gallium oxide. Before accepting this manuscript, the following comments should be taken into account:

  1. Buffer layers to increase structural quality, reduce dislocation density were also used for α-Ga2O3 grown https://iopscience.iop.org/article/10.1088/1361-6463/ac962f/meta

Thank you for providing the recent study, the reference has now been added to the Introduction section.

  1. You write: "... we demonstrated an increase of grain size and crystallinity of MOCVD-grown β-Ga2O3 thin films on c-plane sapphire by using a nucleation layer, which helps to reduce the dislocation density in the initial film..." I did not find estimations of dislocation densities in the text of manuscript.

The phrases related to dislocation density were removed from the manuscript and the focus of the manuscript shifted towards the improved grain size and morphology, as was also suggested by Reviewer 1.

  1. I recommend presenting a table for samples A, B and C with a comparison of their structural properties.

A table with characteristic parameters for surface roughness, crystallinity and grain size has now been added to the manuscript.